# A Novel SNP in *EIF2AK4* Gene Is Associated with Thermal Tolerance Traits in Chinese Cattle

**DOI:** 10.3390/ani9060375

**Published:** 2019-06-19

**Authors:** Kaiyue Wang, Yanhong Cao, Yu Rong, Qingqing Ning, Peng Jia, Yongzhen Huang, Xianyong Lan, Ruihua Dang, Hong Chen, Chuzhao Lei

**Affiliations:** 1Key Laboratory of Animal Genetics, Breeding and Reproduction of Shaanxi Province, College of Animal Science and Technology, Northwest A&F University, Yangling 712100, China; wangkaiyue825@126.com (K.W.); 13653268610@163.com (Y.R.); 18392375865@139.com (Q.N.); nwafujp@gmail.com (P.J.); hyzsci@nwafu.edu.cn (Y.H.); lan342@126.com (X.L.); dangruihua@nwsuaf.edu.cn (R.D.); chenhong1212@263.net (H.C.); 2The Animal Husbandry Research Institute of Guangxi Zhuang Autonomous Region, Nanning 53001, China; caoyh610@163.com

**Keywords:** *EIF2AK4* gene, thermal tolerance, Chinese indigenous cattle, genetic variant, environmental parameters

## Abstract

**Simple Summary:**

China harbors two lineages of cattle (*Bos taurus* and *Bos indicus*) that display pronounced geographical distribution differences. Northern Chinese cattle predominantly belong to *B. taurus* and southern Chinese cattle belong to *B. indicus*. Both *B. taurus* and *B. indicus* contribute to the admixture of cattle in central China. Thermal stress induces oxidative stress and DNA damage in mammals. In general, *B. indicus* are more resistant to thermal stress than *B. taurus*. Eukaryotic translation initiation factor 2-alpha kinase 4 (*EIF2AK4*), which pertains to the family of serine–threonine kinase, is a candidate gene for thermal stress. However, the effects of the bovine *EIF2AK4* gene on the thermal tolerance traits of Chinese cattle breeds remain unknown. Our results suggest that a variant of the *EIF2AK4* gene is associated with thermal tolerance traits in Chinese cattle.

**Abstract:**

Eukaryotic translation initiation factor 2-alpha kinase 4 (*EIF2AK4*, also known as *GCN2*), which pertains to the family of serine–threonine kinase, is involved in oxidative stress and DNA damage repair. A missense single-nucleotide polymorphism (SNP) (NC_037337.1 g.35615224 T > G) in exon 6 of the *EIF2AK4* gene which encodes a p.Ile205Ser substitution was observed in the Bovine Genome Variation Database and Selective Signatures (BGVD). The purpose of the current study is to determine the allelic frequency distribution of the locus and analyze its association with thermal tolerance in Chinese indigenous cattle. In our study, the allelic frequency distribution of the missense mutation (NC_037337.1 g.35615224 T > G) in Chinese cattle was analyzed by sequencing 1105 individuals of 37 breeds including 35 Chinese indigenous cattle breeds and two exotic breeds. In particular, association analysis was carried out between the genotypes and three environmental parameters including annual mean temperature (T), relative humidity (RH), and temperature–humidity index (THI). The frequency of the mutant allele G (NC_037337.1 g.35615224 T > G) gradually decreased from the southern cattle groups to the northern cattle groups, whereas the frequency of the wild-type allele T showed an opposite pattern, consistent with the distribution of indicine and taurine cattle in China. In accordance with the association analysis, genotypes were significantly associated with T (*P* < 0.01), RH (*P* < 0.01), and THI (*P* < 0.01), suggesting that the cattle with genotype GG were found in regions with higher T, RH, and THI. Thus, our results suggest that the mutation (NC_037337.1 g.35615224 T > G) of the *EIF2AK4* gene is associated with thermal tolerance traits in Chinese cattle.

## 1. Introduction

Cellular exposure to elevated temperature activates a large number of anomalies in cellular function [1], such as a general inhibition of protein synthesis, protein structure and function defects, and shifts in metabolism. These anomalies cause huge changes in gene expression and protein synthesis, known as thermal stress response [2,3]. A temporal arrest of translation is one immediate response to thermal stress, which involves thermal-induced phosphorylation of the α subunit of eukaryotic translation initiation factor 2-alpha (eIF2α) to inhibit the formation of the translational initiation complex [4].

Four distinct types of eIF2α kinases (*PKR*, *HRI*, *PERK* and *EIF2AK4*) that are regulated independently in response to various cellular stresses have been identified in eukaryotes. Among these, the *EIF2AK4* gene has been identified to phosphorylate eIF2α on serine 51 and rapidly inhibit translation initiation in response to a wide variety of stress-induced signals including heat shock, oxidative stress, and virus infection [5,6]. Originally, the *EIF2AK4* gene was reported and investigated in yeast, in which *EIF2AK4* phosphorylated the eIF2α and was activated under conditions of nutrient deprivation [7]. Furthermore, a body of studies reported that the *EIF2AK4* gene was also activated by other sources of stress that are not directly related to nutrient deprivation in response to specific stress signals such as UV irradiation [8], virus infection [6], and thermal stress [9]. Recently, comparative genome-wide analyses that detected positive selection signals were conducted in Ethiopian and Asian *Bos indicus* cattle populations using the 80K Indicine BeadChip (GeneSeek Genomic Profiler) and using genotypic data of five *Bos taurus* breeds. The results demonstrated that the *EIF2AK4* gene is associated with cellar stress/thermal tolerance and DNA damage repair [10]. However, the genetic variation of the *EIF2AK4* gene and its association with thermal tolerance in Chinese indigenous cattle breeds are still unclear.

China has plentiful ecosystems and abundant cattle resources, including 53 Chinese indigenous cattle breeds, which are reared in specific geographic regions [11,12]. Studies on Y chromosome polymorphism and mitochondrial DNA markers demonstrated that Chinese indigenous cattle originated from *B. taurus* and *B. indicus* and revealed a diminishing south-to-north gradient of *B. indicus* introgression [13]. Population analysis on the basis of their geographic distributions and morphological characters revealed three fundamental groups of cattle, including northern, central, and southern groups [14]. Additionally, cold temperate and tropical zones are extensively distributed in China, which contributes to the remarkable temperature and humidity differences from north to south. Hence, Chinese indigenous cattle breeds with indicine–taurine ratios varying between zero and one and subject to a broad range of climate are a valuable resource to detect single-nucleotide polymorphisms (SNPs) in the *EIF2AK4* gene and investigate the association between SNPs and environmental parameters consisting of annual mean temperature (T), relative humidity (RH), and temperature–humidity index (THI). Our study contributes to a better understanding of the polymorphism of the *EIF2AK4* gene, which can be used to identify thermal tolerance traits in Chinese cattle.

## 2. Materials and Methods

### 2.1. Ethics Statement

The protocols used in this study and for the animals were recognized by the Faculty Animal Policy and Welfare Committee of Northwest A&F University (FAPWC-NWAFU, Protocol number, NWAFAC1008).

### 2.2. Animals, DNA Extraction, and Data Collection

A total of 1105 individuals from 37 breeds, including 35 Chinese indigenous cattle breeds as well as exotic Angus (*B. taurus*) and Burma (pure indicine population), were investigated (Appendix A). To minimize the degree of relationship among individuals, animals were selected according to both pedigree information and the knowledge of local herdsmen.

The genomic DNA of 1105 individuals was extracted from ear tissues by the standard phenol–chloroform method [15]. The DNA content was determined spectrophotometrically and then the genomic DNA was diluted to 10 ng/µL. All DNA samples were kept at −20 °C until use.

Climatic data (T, RH, THI) over the last 30 years for the sampling sites of the 35 indigenous cattle breeds were collected from the Chinese Central Meteorological Office (http://data.cma.cn) (Appendix A) and were used to estimate thermal tolerance traits.

### 2.3. PCR Analysis of the EIF2AK4 Gene in Chinese Cattle

The polymerase chain reaction (PCR) primer was designed based on the bovine *EIF2AK4* sequence (GenBank accession No. NC_037337.1) to amplify the fragment in *EIF2AK4* using the NCBI database (http://www.ncbi.nlm.nih.gov/). The primer was synthesized by Sangon Biotech (Shanghai) Co., Ltd. Primer sequences; the annealing temperature and fragment size are shown in Appendix A.

The PCR protocol was as follows: each 12.5 µL reaction contained 20 ng of genomic DNA, 20 pmol of each primer, 0.25 mM dNTPs, 1× PCR buffer (including 2.5 mM Mg^2+^), and 1.0 U of rTaq DNA polymerase (Takara, Dalian, China). A thermocycling protocol of 5 min at 95 °C, followed by 32 cycles of 30 s at 94 °C, 30 s at 52.5 °C, 30 s at 72 °C, and a final extension step (72 °C for 10 min) was applied. PCR products were visualized on 1% agarose gels. The remaining amplicon fragments were sequenced by Sangon Biotech (Shanghai, China) Co., Ltd. Sequencing results were analyzed with SEQMAN TMⅡv 6.1 (DNASTAR, Inc., Madison, USA).

### 2.4. Statistical Analysis

The individual cattle were divided according to genotype using PCR amplification and directed sequencing, and the distribution of cattle with the various genotypes was summarized. Genotypic and allelic frequencies were calculated based directly on the observed genotypes in the analyzed breeds. Hardy–Weinberg equilibrium (HWE) was tested based on the likelihood ratio for different locus–population combinations, and the number of observed and effective alleles was determined using POPGENE software (Version 1.32) [16]. Gene heterozygosity (He), gene homozygosity (Ho), and Ne (effective number of alleles, reciprocal of homozygosity) were determined using POPGENE software according to Nei’s methods [17]. Polymorphism information content (PIC) was calculated according to the method of Botstein et al. [18].
$$H_{0}\;=\;\sum_{i\;=\;1}^{n}P_{i}^{2}H_{e}=1-\sum_{i\;=\;1}^{n}P_{i}^{2}\mathrm{Ne}=1/\sum_{i\;=\;1}^{n}P_{i}^{2}$$
$$\mathrm{PIC}\;=\;1-\sum_{i\;=\;1}^{m}P_{i}^{2}-\sum_{i\;=\;1}^{m\;-\;1}\sum_{j\;=\;i\;+\;1}^{m}2P_{i}^{2}P_{j}^{2}$$

Temperature–humidity index is an integrated indicator that is often used to determine whether animals are in a state of thermal stress, and, if so, the degree of thermal stress. The classical index used to evaluate the degree of thermal stress is calculated as:

THI = (1.8T + 32) − (0.55 − 0.0055RH) (1.8T − 26)

where T is temperature in Celsius and RH is relative humidity expressed as a percentage [19].

The SPSS 18.0 software (SPSS, Inc, Armonk, US) was used to analyze the relationship between the genotypes and environmental data in Chinese indigenous cattle breeds. The linear model is:

Y_i_ = µ + G_i_ + B_i_ + e_i_
where Y_i_ = the value of T, RH, and THI between 1951 and 1980; µ = the mean value; G_i_ = the fixed effect of the genotypes; B_i_ = the fixed effect of breeds; e_i_ = the random residual effect. An association analysis was carried out to explore the possible interaction between the genotypes and environmental parameters. *P* < 0.05 was defined as the threshold of significance.

## 3. Results

### 3.1. Diversity Analysis

Genetic indices Ho, He, Ne, and PIC of the mutation (NC_037337.1 g.35615224 T > G) in the *EIF2AK4* gene across 37 cattle breeds are given in Appendix A. In this study, the observed He ranged from 0 to 0.4994. Ne ranged from 1.0000 to 1.9975. The minimum and maximum PIC were 0 and 0.3747, respectively. According to the classification of PIC (PIC value < 0.25, low polymorphism; 0.25 < PIC value < 0.5, intermediate polymorphism; and PIC value > 0.5, high polymorphism), all the northern and central populations possessed intermediate polymorphism, whereas the southern populations possessed intermediate polymorphism or low polymorphism in the locus (NC_037337.1 g.35615224 T > G). The results of the χ^2^ test indicated that apart from five populations (Yanbian, Mongolian, Ji’an, Dianzhong, and Wuling), all the populations’ genotypic frequencies showed Hardy–Weinberg equilibrium (HWE: *P* > 0.05) across Chinese indigenous cattle breeds. The χ^2^ test was not calculated for Ji’an cattle due to the limitation of the sample size.

### 3.2. Analysis of Genotypic and Allelic Frequencies

The analysis of genotypic and allelic frequencies of Chinese indigenous cattle breeds as well as Angus and Burma are shown in Appendix A. Three genotypes (TT, GT, and GG) were detected in 1105 individuals. Moreover, all Angus cattle carried wild-type allele T (100%), while Burma cattle predominantly carried mutant allele G (95.92%), which indicates a differentiated allelic frequency for *B. taurus* and *B. indicus* breeds. The frequencies of the mutant G allele were 0.2818, 0.5822, and 0.7366 in the northern, central, and southern groups, respectively. Wild-type allele T was observed at the highest frequency (0.7182) in the northern group and progressively decreased southward. Furthermore, the highest frequency of the G allele was found in the cattle of Southeastern China, an area which harbors the highest temperature compared to other regions, followed by the cattle in Southwestern China. We then explored the geographical distribution of the *EIF2AK4* gene variation (NC_037337.1 g.35615224 T > G) among 35 Chinese indigenous cattle breeds as well as Angus and pure indicine population, as shown in Figure 1. The results indicate that the frequency of the mutant allele G increased from northern groups to southern groups, while the frequency of the wild-type allele T showed a completely opposite pattern.

### 3.3. Associations of EIF2AK4 Variation with Thermal Tolerance Traits in Chinese Cattle Breeds

The results of the association analysis between the novel SNP (NC_037337.1 g.35615224 T > G) and the three environmental parameters (T, RH, and THI) for different breeds in 1022 Chinese indigenous cattle are shown in Table 1. The cattle with GG (48.24%), GT (33.12%), and TT (18.64%) genotypes were significantly associated with T (*P* < 0.01), RH (*P* < 0.01), and THI (*P* < 0.01). Tests of effects of the three environmental parameters on the *EIF2AK4* genotypes indicated that annual average temperature is closely associated with the genotypes (Appendix A).

## 4. Discussion

As thermal stress has a negative effect on meat, milk, and genetic diversity among cattle breeds, there is an urgent need to explore new methods and strategies to ameliorate the performance of livestock. Abundant cellular resources to maintain an optimal internal temperature are used in mammals, and it is therefore essential for their thermal stress response pathways to be firmly regulated. The thermal tolerance response may have become coordinated into translational regulatory pathways via one or several of the eIF2a kinases during the evolution of thermogenesis in mammals, providing an effective means for genetic regulation during the process of thermal stress [20]. The *EIF2AK4* gene plays a compensatory role for heme-regulated inhibitor (*HRI*) following thermal stress in *Schizosaccharomyces pombe* [21] and mouse germ cells [4]. In addition, Berlanga et al. [22] analyzed the protein levels in the cell extracts by Western blotting and demonstrated that the induction of eIF2 phosphorylation observed in the cells that only express the *EIF2AK4* gene was significantly elevated after longer exposure to thermal stress. However, few studies have investigated the association between thermal tolerance and the *EIF2AK4* gene variation of Chinese indigenous cattle breeds.

There are large contrasting ecoregional differences in China—from cold and dry climate in northern parts to the middle agricultural region and the humid subtropical heat in the southern area [23]. The environment has ubiquitous and copious effects on gene–environment interaction and has a significant correlation with animal performance. China harbors two lineages of cattle that display pronounced geographical distribution differences. Northern Chinese cattle predominantly belong to taurine cattle, whereas southern Chinese cattle are morphologically and genetically recognized as indicine cattle [24,25]. Both taurine and indicine cattle contribute to the admixture of cattle in central China [26]. Additionally, southern cattle breeds, which originated in mountainous areas, are generally resistant to thermal stress, while northern cattle breeds are generally cold-resistant [11,27,28]. Therefore, different types of breeds have a significant influence on thermal tolerance.

In the present work, we screened SNPs in the *EIF2AK4* gene in the database of the Bovine Genome Variation Database and Selective Signatures (BGVD) (http://animal.nwsuaf.edu.cn/code/index.php/BosVar) to confirm whether cellular thermal tolerance is associated with polymorphism in the *EIF2AK4* gene. A novel missense mutation (NC_037337.1 g.35615224 T > G) in the *EIF2AK4* gene was found in Chinese indigenous breeds, which was highly conserved in *B*. *indicus* breeds but absent in *B. taurus*. Genotypic diversity is indispensable to species preservation and improvement of performance with respect to selected animals. In this study, Ji’an cattle represented the smallest observed Ne, suggesting a limited pool of Ji’an sires, while Qinchuan cattle displayed the largest Ne, suggesting higher genetic diversity. Moreover, the result of the χ^2^ test showed that only five breeds (Yanbian, Mongolian, Ji’an, Dianzhong, and Wuling) deviated from HWE among the 35 Chinese cattle breeds, which implies significant differences in genotypic and allelic distributions. Artificial selection is an important factor that affects gene equilibrium in domestic animal populations. Zhang et al. reported that Qinchuan and Jinnan cattle were improved by the introduction of European commercial breeds (such as Simmental) to enhance the production capability of the offspring through artificial insemination in the 1970s [12]. Notably, it is inevitable to break the gene equilibrium between its major gene and other linked genes or genetic markers if one quantitative trait is exposed to artificial selection.

Based on the analysis of genotypic and allelic frequencies of the SNP (NC_037337.1 g.35615224 T > G), we demonstrated that the variation of the *EIF2AK4* gene presented significant geographical differences across Chinese indigenous cattle breeds. On the other hand, the G allele was predominant in *B. indicus* (Burma) and the T allele was predominant in *B. taurus* (Angus). Similar patterns existed in the Chinese indigenous cattle breeds (Figure 1). The results demonstrate that the SNP (NC_037337.1 g.35615224 T > G) presented a clear geographical distribution across Chinese breeds of cattle. Taken together, these results show that the frequency of the mutant G allele gradually decreased from the southern region to the northern region of China, while the wild-type T allele showed an opposite pattern, consistent with the distribution of indicine and taurine cattle in China.

The association between the novel SNP (NC_037337.1 g.35615224 T > G) and the three environmental parameters (T, RH, and THI) showed that the GG genotype had a greater frequency in regions with higher T, RH, and THI. Moreover, all three genotypes at the NC_037337.1 g.35615224 T > G locus may be associated with thermal tolerance. With the availability of high-density single-nucleotide polymorphisms over the last few years, some insightful studies have been performed to conduct association analysis and identify SNPs as genetic markers of selection for economic characters. For instance, Xu et al. [29] reported that an indel maker of the *PLAG1* gene can be used as a candidate molecular marker for breeding in cattle. Our study suggested that variation of the bovine *EIF2AK4* gene may affect thermal tolerance. However, the results need to be validated further by testing animals with different genotypes and recording their performance and physiological response.

In fact, earlier reports found that exposure to a specific environmental stress would promote new robust genotypes that are resistant to environmental stress [30]. In addition, several studies have supported the concept that pronounced introgression among different bovine species (including yak, gayal, gaur, and banteng) might facilitate adaptation to local environments [31,32]. Finally, to improve economic performance, China has been introducing exotic beef and dairy breeds to improve indigenous cattle since the 1970s [33]. These factors may help us elucidate a few mutant G alleles and wild-type T alleles in the northern and southern regions, respectively. Thus, our results suggest that the SNP (NC_037337.1 g.35615224 T > G) of the *EIF2AK4* gene is associated with thermal tolerance traits in Chinese cattle, as also shown in previous reports [4,21,34].

## 5. Conclusions

By comparing *B. indicus* and *B. taurus*, the genetic diversity of the missense mutation (NC_037337.1 g.35615224 T > G) in the *EIF2AK4* gene was detected. The relationship between the *EIF2AK4* gene and thermal tolerance traits in Chinese cattle could be validated further by testing animals with different genotypes and recording their performance and physiological response.

## Figures and Tables

**Figure 1 animals-09-00375-f001:**
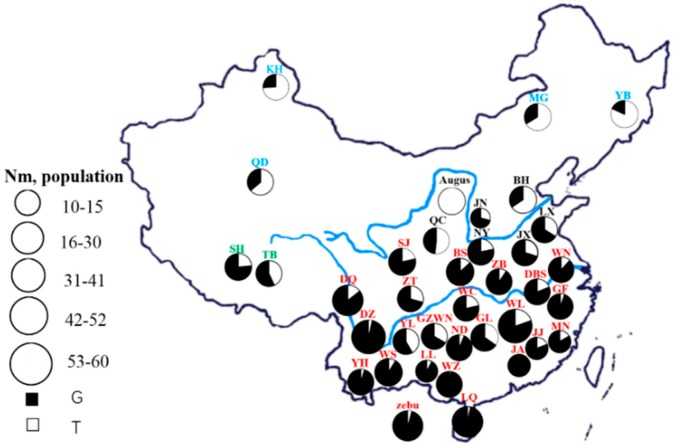
Geographical distribution of G and T alleles of the locus (NC_037337.1 g.35615224 T > G) of the Eukaryotic translation initiation factor 2-alpha kinase 4 (*EIF2AK4*) gene among 37 cattle breeds in China, including 35 Chinese indigenous cattle breeds as well as Angus and pure indicine population. Circled areas are proportional to the sample size. Blue, black, red, and green fonts represent northern, central, southern, and special groups, respectively. BH, Bohai Black; BS, Bashan; DBS, Dabieshan; DZ, Dianzhong; DQ, Diqing; GF, Guangfeng; GL, Guanling; GZWN, Weining; JA, Ji’an; JJ, Jinjiang; JN, Jinnan; JX, Jiaxian red; KZ, Kazakh; LQ, Leiqiong; LX, Luxi; LL, Longlin; MG, Mongolian; MN, Minnan; NY, Nanyang; ND, Nandan; QC, Qinchuan; QD, Qaidam; SJ, Sanjiang; SH, Shigatse Humped; TB, Tibetan; WC, Wuchuan; WL, Wuling; WN, Wannan; WS, Wenshan; WZ, Weizhou; YB, Yanbian; YL, Yunling; YH, Yunnan Humped; ZB, Zaobei; ZT, Zhaotong; zebu, pure indicine population.

**Table 1 animals-09-00375-t001:** Least squares mean and standard error for the temperature (T), relative humidity (RH), and temperature–humidity index (THI) of different genotypes of the single-nucleotide polymorphism (SNP) (NC_037337.1 g.35615224 T > G) in the *EIF2AK4* gene.

SNP	Genotype (n)	Temperature	Relative Humidity	Temperature–Humidity
		(°C) (LSM ± SE)	(%) (LSM ± SE)	Index (LSM ± SE)
*EIF2AK4:* NC_037337.1: g.35615224 T > G	TT (121)	9.016^A^ ± 0.3487	62.16^A^ ± 1.094	50.4976^A^ ± 0.47930
	GT (310)	11.113^B^ ± 0.2914	65.64^B^ ± 0.706	53.4225^B^ ± 0.40590
	GG (591)	15.072^C^ ± 0.2471	73.68^C^ ± 0.392	58.8280^C^ ± 0.35644

LSM ± SE, the least square means with standard errors for diverse genotypes and environment parameters. Means in the same column and locus with difference capital superscripts, A, B and C, are different at *P* < 0.01.

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
