# Peer review of "A Novel SNP in EIF2AK4 Gene Is Associated with Thermal Tolerance Traits in Chinese Cattle"

_animals, 2019, doi:10.3390/ani9060375_

Round 1
Reviewer 1 Report
Journal: animals
Title: A novel SNP in is associated with thermal tolerance traits in Chinese cattle Article
Type: Research Paper
Reviewer Comments:
The strength of this manuscript is that it highlighted the role of single nucleotide polymorphism (T/G) in the EIF2AK4 gene in response to temperature stress tolerance in Chinese cattle. It has used a robust sample of 1105 individuals from 37 breeds (35 Chinese breeds and two exotic breeds). The protocols used for making the present inferences are established, standard, and repeatable. The weakness of this manuscript is the present investigation is preliminary work and it needs further research to validate the present findings.
Question 1: What is south to north gradient? Not clear. Revise this statement.
Question 2: Were 1105 individuals from 37 breeds used for the present investigation? If so, then why in the abstract section (Line 33) “1022 individuals of 35 populations were used for sequencing purpose”)?
Question 3: Line 148: What does the authors mean by “Ne value were approaching 1.9975”? Explain it.
Question 4: Line 1163: What does the author mean by “showing characterize of”? Explain it.
Question 5: As the experiments were carried with cattle. Why no ethical statement was included by the authors in the manuscript?
Question 6: Why the authors have not mentioned the temperature/average temperature of southern and northern region of China, which can clearly show the interpretation and valuable for other readers for doing further comparison and experimentations.
Question 7: There are some grammatical errors that need to be corrected.
For example, Line 43: “strikingly geographical distribution”
Modify to: “stinking geographic distribution”
Line 45: “In according with”
Modify to “in accordance with”
Author Response
Response to Reviewer 1 Comments
Point 1: What is south to north gradient? Not clear. Revise this statement.
Response 1: Thanks. We have revised the statement as follows: “Studies on Y chromosome polymorphism and mitochondrial DNA markers demonstrated that Chinese indigenous cattle originated from Bos taurus and Bos indicus and revealed a diminishing south-to-north gradient of Bos indicus introgression [Cai, X., et al., mtDNA diversity and genetic lineages of eighteen cattle breeds from Bos taurus and Bos indicus in China. Genetica, 2007. 131(2): p. 175-183.].” (lines 80-83)
Point 2: Were 1105 individuals from 37 breeds used for the present investigation? If so, then why in the abstract section (Line 33) “1022 individuals of 35 populations were used for sequencing purpose”)?
Response 2: We are extremely grateful to you for pointing out this problem, and the sentence has been corrected. “In our study, the genetic variation of the missense SNP (NC_037337.1 g.35615224 T>G) in Chinese cattle were analyzed by sequencing from 1105 individuals of 37 breeds including 35 Chinese indigenous cattle breeds and two exotic breeds.” (lines 38-41)
Point 3: Line 148: What does the authors mean by “Ne value were approaching 1.9975”? Explain it.
Response 3: Thanks. We have revised it to avoid ambiguous statement as follows: “In this study, the observed He values ranged from 0 to 0.4994. Ne values ranged from 1.0000 to 1.9975.” (lines 156-157)
Point 4: Line 163: What does the author mean by “showing characterize of”? Explain it.
Response 4: Allelic frequencies of Angus and Burma partially reflect the indicine–taurine characters.
We admit that there is a problem with my own expression leading to misinterpret the meaning, and the sentence has been revised to make the meaning of the expression more accurate. “Moreover, we detected that all Angus cattle carried wild-type allele T (100%) and Burma cattle predominantly carried mutant allele G (95.92%) using a relatively higher density and Bos indicus derived SNP, showing characterize on allelic frequency for Bos taurus and Bos indicus breeds to some extent.” (lines 170-174).
Point 5: As the experiments were carried with cattle. Why no ethical statement was included by the authors in the manuscript?
Response 5: We are very sorry for our neglect of ethic information and have added ethical statement. “The protocols used in this study and for the animals were recognized by the Administration of Affairs Concerning Experiment Animals (Ministry of Science and Technology, China, 2004) and the Faculty Animal Policy and Welfare Committee of Northwest A&F University (FAPWC-NWAFU).” (lines 96-99)
Point 6: Why the authors have not mentioned the temperature/average temperature of southern and northern region of China, which can clearly show the interpretation and valuable for other readers for doing further comparison and experimentations.
Response 6: We have mentioned the mean annual temperature, mean relative humidity and mean temperature humidity index as a reference, but choose not to devote more detail information to discuss it, given the prior work. “China has extremely contrasting ecoregional differences: from the northern cold and dry climate to the middle agriculture region and the humid subtropical heat in the southern area [Zeng, L., et al., PRLH and SOD 1 gene variations associated with heat tolerance in Chinese cattle. Animal Genetics, 2018. 49(5): p. 447-451.].” (lines 230-231)
Point 7: There are some grammatical errors that need to be corrected.
Response 7: We regret there were problems with the grammar. The paper has been carefully revised to improve the grammar and readability. Such as “In accordance with” “striking geographical distribution”

Reviewer 2 Report
The article has two main objectives: a) study the allele and genotype frequency of EIF2AK4 gene in 37 breeds and b) to study the association between SNP polymorphism and heat tolerance indicators (temperature, humidity and THI). The authors’ conclusion and approach is valid for the first objective but I have two main concerns as below on their second objective:
1- In the data analysis section either I am not clear or they failed to explained very well how they considered SNP polymorphism in the model? How did they include breed effect? Have they tried breed as fixed versus random effect in the model? Have you accounted for different N of breeds?
2- Why do they use 30 years weather info? And why from 1951 to 1980? Is 30 years period enough to see the population change and the association?
Overall, I do not think you can make the recommendation that this SNP could be used as a maker for selecting animals for heat tolerance, but they could make recommendation such that other studies may take place using their results and testing animals with different genotypes and recording their performance and physiological response. With the current STAT analysis, I do not think they can make claim of relationship between SNP polymorphism and environment indicators.
Here are the minor comments in text:
L27, you need to have one line about your results then you can make conclusion as you did in L27 “The results provide….”
“L79, would you use some firm numbers from reference(s) rather than using approximately.
L95, incomplete sentence
L140 In the Linear model I do not see where they have fitted the polymorphism effect?
L120, comma before and the ..
L122, (HWE) has been …
L137, what does correctly mean here?
L142, The least …. I am not clear why do you want to say here. What would be the objective of in this sentence
L143 Data expressed in where ?
L149, should not be in the other order your Max and Min
L233, It has been suggested that 5 breeds were not in HWE due to AI. But would you provide some numbers on the rate of AI to back up your claim. I would also use term may be
Author Response
Response to Reviewer 2 Comments
Point 1: In the data analysis section either I am not clear or they failed to explained very well how they considered SNP polymorphism in the model? How did they include breed effect? Have they tried breed as fixed versus random effect in the model? Have you accounted for different N of breeds?
Response 1: Thanks.
(1) We considered different genotypes as fixed effect in the model.
(2) Firstly, we include breed effect in the model. Association analyses were carried out to explore the possible interaction combing genotypes and environmental parameter, whereas breed effect is not a research focus. We have revised the statistical analyze section (line 147) to include breed information, taking into account your critique. Secondly, we have added in the discussion: “Southern cattle breeds, which originated in mountainous areas, are generally resistant to thermal, while northern cattle breeds are generally cold resistance.” “Therefore, breed factor has a significant relationship with heat resistance.” Finally, the model was similar to that of single marker association analysis. [Huang Y Z, Zhang E P, Chen H, et al. Novel 12-bp deletion in the coding region of the bovineNPM1 gene affects growth traits[J].] [Journal of applied genetics, 2010, 51(2): 199-202. Zeng, L., et al., PRLH and SOD 1 gene variations associated with heat tolerance in Chinese cattle. Animal Genetics, 2018. 49(5): p. 447-451.]
(3) We agree that the association of breeds with environmental parameters is important, and we tried breed as fixed effect in the model.
(4)Yes, I have accounted for different N of breeds. And we found that breed factor has a significant relationship with T, RH and THI, which is parallel to previous research results (Southern cattle breeds are more resistant to thermal stress compared with northern cattle breeds).[Huai, Q., J. Zhiyong, and C. Zhijie, A survey of cattle production in China. World Animal Review, 1993. 76.][ Dolberg, F., Progress in the utilization of urea-ammonia treated crop residues: biological and socio-economic aspects of animal production and application of the technology on small farms. Livestock Research for Rural Development, 1992. 4(2): p. 20-32.] [Dolberg, F. and P. Finlayson, Treated straw for beef production in China. World Animal Review, 1995. 82(1): p. 14.]
Point 2: Why do they use 30 years weather info? And why from 1951 to 1980? Is 30 years period enough to see the population change and the association?
Response 2: We have tried to collect more weather information in China, while we can only collect the complete weather information for these 30 years in China at present. Additionally, a 30-year whether information was also used by Blackburn et al. [Blackburn H D, Krehbiel B, Ericsson S A, et al. A fine structure genetic analysis evaluating ecoregional adaptability of a Bos taurus breed (Hereford)[J]. PloS one, 2017, 12(5): e0176474.]
Point 3: Overall, I do not think you can make the recommendation that this SNP could be used as a maker for selecting animals for heat tolerance, but they could make recommendation such that other studies may take place using their results and testing animals with different genotypes and recording their performance and physiological response. With the current STAT analysis, I do not think they can make claim of relationship between SNP polymorphism and environment indicators.
Response 3: Thank you. The conclusions drawn from the statistical analysis are corrected as you suggested: “In total, by comparing Bos indicus and Bos taurus cattle breeds, the missense mutation within EIF2AK4 gene was detected in this study. The variant of the EIF2AK4 gene correlated with thermal tolerant traits in Chinese cattle could be further validated via testing animals with different genotypes and recording their performance and physiological response.” (lines 294–297) Additionally, the corresponding changes have been made elsewhere in the revised manuscript.
Point 4: L27, you need to have one line about your results then you can make conclusion as you did in L27 “The results provide….”
Response 4: We agree with the reviewer and have added the following sentences in the simple summary: “Our result may suggest that variant of the EIF2AK4 gene is correlated with thermal tolerant traits in Chinese cattle. The results of this study need to be further validated via testing animals with different genotypes and recording their performance and physiological response.” (lines 26-29)
Point 5: L79, would you use some firm numbers from reference(s) rather than using approximately.
Response 5: Thank you for this valuable feedback. We agree with the reviewer and have corrected the sentence. “China has plentiful ecosystem and abundant cattle resources, including 53 Chinese indigenous cattle breeds, which were reared in specific geographic regions.” (lines 79-80)
Point 6: L95, incomplete sentence
Response 6: Thank you so much! The sentence has been corrected. “A total of 1105 individuals from 37 breeds including 35 Chinese indigenous cattle breeds as well as exotic Angus (Bos taurus) and Burma (Bos indicus) were investigated.” (lines 100–101)
Point 7: L140 In the Linear model I do not see where they have fitted the polymorphism effect?
Response 7: In the model, Yi is the value of T, RH, THI between 1951-1980; µ is the mean value; Gi and Bi are the fixed effect of the genotypes and breeds, respectively; ei is the random residual effect. We can use the model to analyze the association between genotypes and environmental parameters.
Point 8: L120, comma before and the
Response 8: Thanks. We have added a comma as you suggested. “The different individuals of cattle were divided into various genotypes using PCR amplify and directed sequencing, and the distribution of different types with cattle was summarized.” (lines 126-127)
Point 9: L122, (HWE) has been …
Response 9: The correction has been made. “Hardy-Weinberg equilibrium (HWE) was tested based on likelihood ratio for different locus-population combinations and the number of observed and effective alleles by POPGENE software (Version 1.32).” (lines 129-131)
Point 10: L137, what does correctly mean here?
Response 10: Maybe we used an inappropriate adverb, while we just want to express “reasonably” “precisely”. The confused word was detected in the revised version. “To determine the relationship between the polymorphism of the EIF2AK4 gene and thermal tolerance traits, the correlation between environmental data and genotypes in Chinese indigenous cattle breeds was analyzed using the SPSS 18.0 software (SPSS, Inc).” (lines 143-145)
Point 11: L142, The least …. I am not clear why do you want to say here. What would be the objective of in this sentence?
Response 11: We are very sorry for our incorrect writing and have removed the information to (lines 209-210).
Point 12: L143 Data expressed in where?
Response 12: We are very sorry for our incorrect writing and have removed the information to line 210.
Point 13: L149, should not be in the other order your Max and Min
Response 13: We have made correction according to the reviewer’ comment. “The minimum and maximum of PIC values were 0 and 0.3747, respectively.” (lines 157-158).
Point 14: L233, it has been suggested that 5 breeds were not in HWE due to AI. But would you provide some numbers on the rate of AI to back up your claim. I would also use term may be
Response 14: In the revised version, the paragraph containing above text has been rewritten as reviewer suggested. “Zhang et al. reported that Qinchuan and Jinnan cattle were improved by introduction of European commercial breeds blood (such as Simmental) to enhance the production capability of offspring through artificial insemination in 1970s [12]. Notably, it is inevitable to break the gene equilibrium between its major gene and other linked genes or genetic markers.” (lines 255-259).

Round 2
Reviewer 2 Report
No more comments.
Author Response
Thanks for your valuable comments on our paper.